# Physical, Rheological, and Permanent Deformation Behaviors of WMA-RAP Asphalt Binders

Kátia Aline Bohn [1,*], Liseane Padilha Thives [1] and Luciano Pivoto Specht [2]

1 Postgraduate Program in Civil Engineering, Federal University of Santa Catarina, Florianópolis 88040-900, Brazil; liseane.thives@ufsc.br
2 Department of Transportation, Federal University of Santa Maria, Santa Maria 97105-900, Brazil; luspecht@ufsm.br
* Correspondence: katia.bohn@posgrad.ufsc.br

**Abstract:** With the rapid global expansion of road networks, the asphalt industry faces several environmental challenges, such as material shortages, environmental concerns, escalating material costs, demand for eco-friendly materials, and the implementation of "Net Zero" policies. Given these challenges and recognizing the need to explore new solutions, this research evaluated asphalt binder samples incorporating Warm Mix Asphalt (WMA) and Reclaimed Asphalt Pavement (RAP), or WMA-RAP. The assessment focused on analyzing the physical, rheological, and permanent deformation characteristics of WMA-RAP samples containing 20%, 35%, and 50% recycled pavement. The study utilized a chemical surfactant-type WMA additive, Evotherm® P25. The findings showed that the WMA-RAP combination resulted in increased stiffness ranging from 247% to 380% and a reduced phase angle of 16% to 26% with an increasing RAP content from 20% to 50% at $T_{ref}$ 20 °C and 10 Hz. Furthermore, the penetration decreased from 20% to 47%, and the softening point increased from 7% to 17%. An improvement of 2 PGHs was observed by adding 35% and 50% RAP. Additionally, WMA samples containing up to 50% RAP presented more elevated permanent deformation resistance, supporting traffic levels of 64V or 70H. WMA-RAP binders allow mixture production at lower temperatures—an amount of 30 °C less—conserving energy and decreasing the need for new aggregate materials by incorporating recycled materials, thus minimizing the environmental impact.

**Keywords:** warm asphalt binder; reclaimed asphalt pavement; WMA-RAP; industrial sustainability materials; recycling materials; permanent deformation; MSCR

## 1. Introduction

Rapid economic growth, population expansion, and increased industrial production have historically relied on resource depletion, resulting in environmental issues such as pollution and global warming. Recycling and reusing waste materials offer potential solutions to reduce resource dependence and pollution. Given the enormous global demand for asphalt in road construction, Reclaimed Asphalt Pavement (RAP) has gained significant attention as an environmentally friendly alternative. Researchers have been increasingly exploring the incorporation of RAP into asphalt mixes to enhance sustainability and reduce environmental impact.

During a highway's service life, pavement restoration interventions are inevitable, leading to a substantial generation of RAP. Pavement recycling is a technique that repurposes RAP in new mixtures, as this material possesses significant economic value due to its aged asphalt binder while concurrently reducing the demand for new aggregates, thus conserving natural resources [1]. Hence, studying methods and alternatives that incorporate RAP into new pavements without compromising their performance becomes imperative, given that aged binders tend to lose their properties due to oxidation, potentially affecting their behavior in conjunction with other materials.

Some research on RAP's physical, rheological, and predictive behavior has yielded results that advance state-of-the-art knowledge of recycled pavement performance [2–5]. The asphalt binder extracted from RAP exhibits distinct chemical and rheological properties compared to virgin asphalt binders [6]. Consequently, specific concerns arise regarding RAP utilization: the selection of base binder, shifts in mixing and compaction temperatures, susceptibility to temperature, resistance to permanent deformation, fatigue performance, and chemical alterations [1].

Studies [7,8] have demonstrated that asphalt mixtures containing RAP can achieve performance equivalent to conventional Hot Mix Asphalt (HMA) or even superior mechanical performance [9]. Recycled pavement encompasses both aged and rigid asphalt binder, and thus, the inclusion of RAP in new mixtures augments resistance to permanent deformation [2,10–12]. In investigating the viscoelastic behavior of recycled mixtures, Zheng et al. [13] observed an increase in the elastic component with higher RAP content and loading frequency, enhancing stability at elevated temperatures. Furthermore, dynamic modulus values are augmented with decreasing temperature and increasing frequency [13]. However, RAP within HMA mixtures could potentially experience secondary aging during high-temperature mixing stages.

Further research indicates that the successful production of asphalt pavements incorporating RAP necessitates carefully selecting milling types and production processes for the mixtures [14,15]. Apart from reducing the consumption of virgin materials, RAP incorporation also reduces the production cost of asphalt mixtures compared to conventional blends [16].

Given the global prominence of "Net Zero" policies in recent years, driven by climate concerns and the need to reduce greenhouse gas emissions to limit global warming to 1.5 degrees Celsius above pre-industrial levels, the field of pavement engineering has explored more sustainable technologies. These policies aim to balance emissions with their removal, making them crucial in climate change minimization [17–19]. One such technology is Warm Mix Asphalt (WMA), which reduces sample manufacturing and compaction temperatures, thus conferring environmental benefits such as reduced energy consumption during production and, consequently, diminished fuel usage and emissions [20–22].

At the same time, with this technology, the focus on recycling gains prominence, as the lower heating temperature of WMA enables higher RAP content [23–27], mitigating further degradation of the aged binder in RAP. The potential to incorporate a greater amount of RAP is due to the reduced mixing temperature, resulting in less binder aging and assisting in counteracting the aging of the old binder from the recycled material [27]. Research indicates that RAP can enhance the performance of WMA mixtures compared to HMA [28–31].

Utilizing WMA substantially reduces fume exposure during pavement operations compared to HMA [26,32,33], with a fume reduction of about 50% for every 12 °C reduction in temperature [33]. Additionally, the energy required during the compaction process is diminished, aiding in greenhouse gas emission reduction. A 30 °C reduction in aggregate heating temperature leads to a 15% or more reduction in fuel costs for mixture production, translating to a threefold reduction in emissions [34]. Compared to a new asphalt mixture comprising virgin aggregates and binders, the cost of asphalt mixtures with 50% RAP decreases by over 30%, and WMA additives reduce energy consumption by around 20% [30].

Typically, mixtures with high RAP content consist of 20% to 30% incorporated recycled material, contingent on the country and available milling type [35], with the vast majority applied in HMA mixtures. Thus, the common practice involves employing 25% RAP [36]. Some studies evaluating the feasibility of incorporating high RAP contents into warm mixtures suggest that WMA technologies could be used with 75% RAP [37], 90 to 100% RAP [26], and 100% RAP [38], thereby enhancing the workability of the new asphalt mixture. Enhanced workability is a key feature of the WMA additive [25], and it is also applicable to mixtures without RAP. More documented information is needed

regarding the utilization of RAP in WMA mixtures compared to conventional ones, as well as data on the long-term field performance with high proportions of recycled pavement, which is necessary to investigate their durability throughout their service life.

Numerous studies have explored the performance of WMA-RAP mixtures, encompassing different RAP proportions and various WMA additives [37,39–43]. The literature highlights that an increase in RAP within WMA enhances the resistance to permanent deformation of mixtures [28,30,37,39–44] due to the heightened rigidity of the aged binder derived from RAP.

However, the resistance to permanent deformation of WMA-RAP mixtures remains a concern, as Zhao et al. [42] revealed a greater rutting depth than HMA mixtures. Nevertheless, Kim et al. [28] indicated that the field performance of RAP-containing mixtures after 5 to 6 years of service demonstrated favorable resistance to permanent deformation, with mean rutting depths below 6 mm.

The studies predominantly underscored the same benefits of WMA-RAP, highlighting its significant environmental and economic potential. Reducing production and compaction temperatures of asphalt mixtures through WMA additives [20–22] makes the process ecologically sound and sustainable, enabling more significant RAP usage [25,29] and yielding promising performance outcomes for new mixtures [24,30,41].

Considering the viscoelastic behavior, which depends on temperature, time, and load application rate [45–47], alterations in the physical and rheological properties of asphalt materials can arise due to adding additives and aged binders. Hence, it is crucial to initially investigate the type of RAP to be employed and its interaction with virgin binder and WMA additives to ascertain the proportion of aged binder that can be incorporated into these materials without compromising the final sample's physical, rheological, and performance properties. The main challenge remains the extraction of aged asphalt binder from RAP [48] for more in-depth studies concerning its interaction with other materials.

This study evaluated the utilization of WMA-RAP asphalt binders, incorporating different proportions of RAP, to evaluate their physical, rheological, and permanent deformation performance behaviors. Focusing on the asphalt binder level of analysis is justified as it helps explore and develop higher-quality mixtures. Various factors influence the characteristics and performance properties of asphalt materials when combined with additives and other asphalt binders. Consequently, the study compared the WMA-RAP samples with reference samples, including HMA, WMA, and the original RAP prototype, to assess potential shifts in behavior. The objective was to determine how the increased proportion of aged RAP material affects sample properties and whether the use of WMA can mitigate the accelerated aging observed in materials during new mixture production. Additionally, the study aimed to identify an optimal RAP percentage based on the findings.

## 2. Materials and Methods

### 2.1. Materials

The materials employed in the research comprised conventional unmodified virgin asphalt (Neat Binder, Curitiba, Brazil), Warm Mix Asphalt (WMA) additive, and Reclaimed Asphalt Pavement (RAP) binder (Neat RAP Binder).

The RAP originated from the milling of a surface course with approximately 2 years of micro-coating over a layer with 9 years of service, from a federal highway in the southern region of Brazil, BR-287, between the cities of Mata and São Vicente do Sul, in Rio Grande do Sul State.

The asphalt binder used was conventional unmodified, classified by penetration grade of 50/70. Results of the asphalt binder characterization are presented in this research. WMA additive incorporated was of the chemical surfactant type, Evotherm® P25 (Ingevity–North Charleston/SC/USA). Figure 1 presents the materials utilized. In Table 1 are indicated the physical and chemical properties of Evotherm® P25.

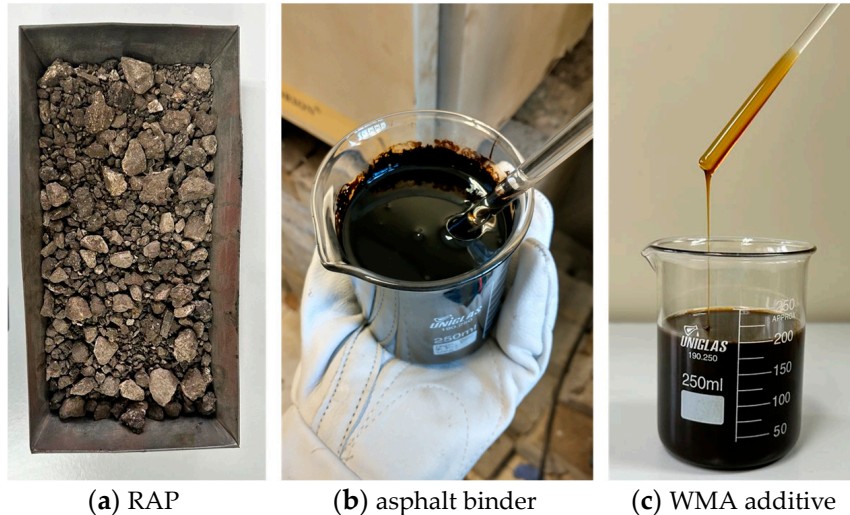

|            |                |                |
|:----------:|:--------------:|:--------------:|
| (**a**) RAP | (**b**) asphalt binder | (**c**) WMA additive |

**Figure 1.** Research materials.

**Table 1.** Evotherm® P25 properties.

| Physical State | Liquid |
|:--------------:|:------:|
| Color | Brown |
| Odor | Mild |
| pH | 2.3 [Conc. (% *w/w*): 5%] (in 50:50 IPA/Water) |
| Melting Point | $<-16\ °C$ |
| Boiling Point | $>100\ °C$ |
| Flash Point | Closed Cup: 181 °C |
| Relative Density | 0.99 |
| Solubility | Insoluble in cold and hot water |
| Viscosity | Dynamic (room temperature): 487 cP |

Source: Data adapted from supplier [49].

### 2.2. Method

Initially, the asphalt binder was extracted from Reclaimed Asphalt Pavement (RAP) using a rotary evaporator and trichloro-ethylene (TCE) as the solvent, according to the standard BS EN 12697-3+A1 [50]. The extraction process consisted of two phases:

- The first phase was conducted at a temperature of 90 ± 5 °C and a pressure of 40 ± 5 kPa. During this phase, the solution (solvent + asphalt binder) was transferred to the evaporation flask.
- The second phase was carried out at a temperature of 160 ± 5 °C and a pressure of 2 ± 0.5 kPa to remove any residual solvent from the sample. The process was considered complete when no more bubbles were observed forming in the sample within the evaporation flask.

The results of the RAP binder characterization are presented in this research.

Relative to the total asphalt binder content of the sample, increments of 0.5% of Warm Mix Asphalt (WMA) additive were added. A total of six asphalt binder samples were evaluated. For comparison, two reference samples without RAP were tested: one hot (HMA_Ref) and one warm (WMA_Ref). Additionally, the sample consisting solely of RAP binder (RAP_Original) was also evaluated. The WMA-RAP samples were tested with 20%, 35%, and 50% RAP incorporation rates, designated as "RAP", followed by the incorporated RAP content in percentage. The compositions of the studied samples are presented in Table 2.

**Table 2.** Composition of the evaluated asphalt binder samples.

| Asphalt Binder Nomenclature | Sample Composition | |
|---|---|---|
| | % Asphalt Binder | % RAP |
| HMA_Ref | 100 | 0 |
| WMA_Ref | 100 | 0 |
| RAP_20% | 80 | 20 |
| RAP_35% | 65 | 35 |
| RAP_50% | 50 | 50 |
| RAP_Original | 0 | 100 |

The sample preparation protocol followed the guidelines of ASTM D4887 [51]. The asphalt binders were heated to a maximum temperature of 135 °C. In a glass beaker, proportions of conventional asphalt binder, RAP, and WMA additive were added and homogenized using a glass rod. The sample was maintained in the oven at 135 °C for 10 min and, after 5 min, mixed using a glass rod for 30 s. After this process, the sample was ready to perform the tests.

The samples were characterized and evaluated in their virgin/original condition and after short-term aging. The binders were aged using the Rolling Thin Film Oven (RTFO) equipment, following the ASTM D2872 [52] guidelines. The RTFO test was conducted at 133 °C for the additive-modified samples, which was 30 °C lower than the samples without additive, as per the manufacturer's recommendation for temperature reduction when working with WMA additives.

To achieve the study objective of assessing the behavior of WMA-RAP samples in comparison to reference samples with varying RAP contents, with a focus on physical, rheological, and performance characterization related to permanent deformation, the conducted tests are detailed in Table 3.

**Table 3.** Tests and evaluated properties for the characterization of the samples and their respective standards.

| Properties/Tests | Standard |
|---|---|
| Penetration | ASTM D5 [53] |
| Softening Point | ASTM D36 [54] |
| Viscosity (135 °C, 150 °C e 177 °C) | ASTM D4402 [55] |
| Complex Modulus (G*) and Phase Angle (δ)–PGH and Master Curves | ASTM D7175 [56], AASHTO M320 [57] and AASHTO M332 [58] |
| Elastic Recovery (%R) e Non-Recoverable Creep Compliance ($J_{nr}$)-MSCR | ASTM D7405 [59] and AASHTO T350 [60] |

Furthermore, the study aimed to evaluate how aging could impact the behavior of these samples and determine the optimal percentage of RAP to provide high-temperature performance pavements. Figure 2 illustrates the research flowchart.

### 2.2.1. High-Temperature Performance Grade

The High-Temperature Performance Grade (PGH) and continuous PGH were determined by analyzing the data obtained from the Dynamic Shear Rheometer (DSR), including the Dynamic Shear Modulus ($|G^*|$) and Phase Angle (δ). The controlled deformation test followed the stop criteria and deformation values outlined in Table 4. Furthermore, the PGH was also determined using data from the Multi-Stress Creep and Recovery Test (MSCR) to establish the level of traffic stress absorbed by each asphalt binder.

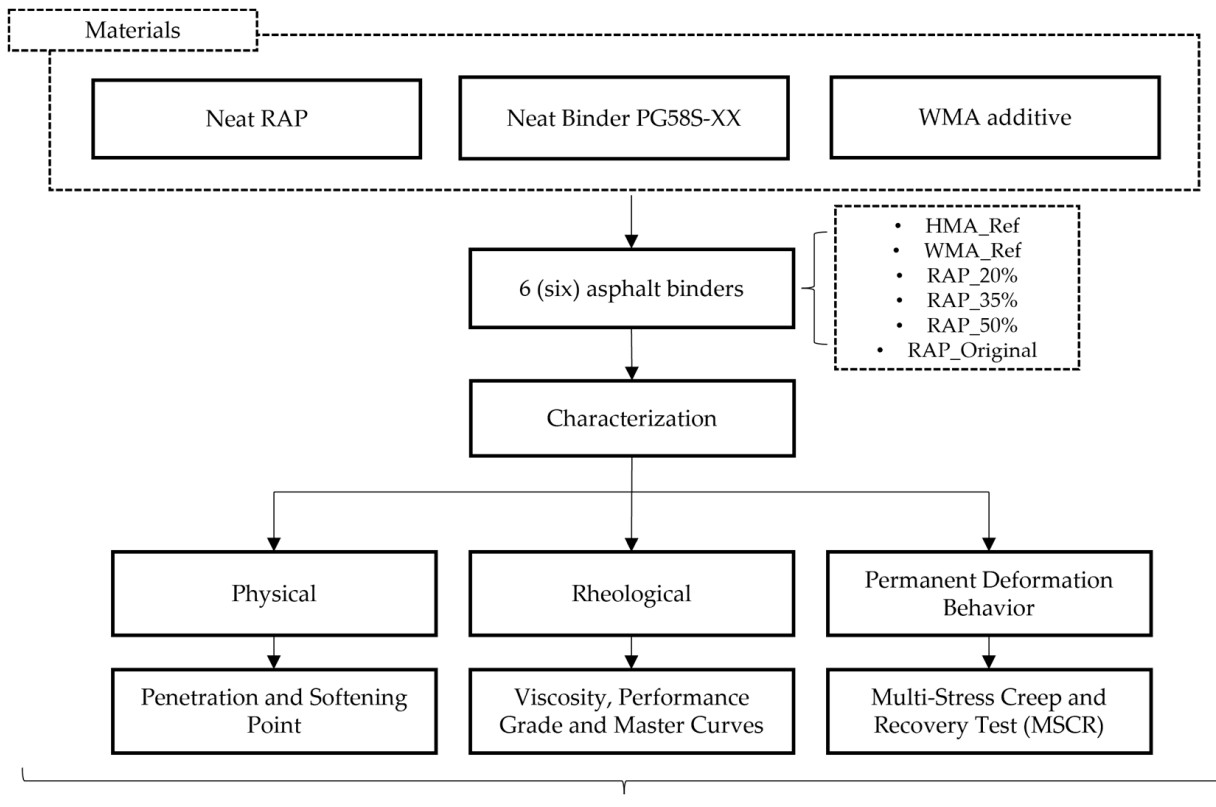

**Figure 2.** Experimental flowchart of the asphalt binders analyzed.

**Table 4.** Stop criteria and controlled deformation percentage used in the test.

| Sample | G*/Senδ (kPa) | Deformation (%) |
|---|---|---|
| Virgin/Original Binder | 1.0 | 12 |
| RTFO Binder | 2.2 | 10 |

Source: Adapted from ASTM D7175 [56].

### 2.2.2. Master Curves

The materials were also subjected to a frequency sweep in a linear ramp, ranging from 0.1 to 30 Hz, and a temperature sweep from 5 °C to 75 °C, which provided data for |G*| and δ. The test was conducted at 5 °C, 20 °C, and 35 °C using an 8 mm geometry and a 2 mm gap. For temperatures of 35 °C, 60 °C, and 75 °C, a 25 mm geometry and a 1 mm gap were employed. The test was carried out under a controlled deformation of 1%.

The δ value corresponds to the elastic-to-viscous response ratio during the shear process, indicating the relative amount of recoverable and non-recoverable deformation. The |G*| is a measure of the material's overall resistance to deformation when subjected to repeated pulses of shear deformation. Both parameters reflect the linear viscoelastic characterization of the asphalt binders.

With the material responses under different temperatures and loading frequencies, master curves of |G*| and δ were constructed using the Time-Temperature Superposition Principle (TTSP) for a reference temperature ($T_{ref}$) of 20 °C. The shifting factors ($a_T$) were

obtained using Equation (1) and adjusted using the Williams-Landel-Ferry (WLF) model, as shown in Equation (2).

$$a_T(T) = \frac{f(T)}{f\left(T_{ref}\right)} \qquad \rightarrow \qquad a_T\left(T_{ref}\right) = 1, \tag{1}$$

$$\log(a_T) = -\frac{C_1 \times \left(T - T_{ref}\right)}{C_2 + T - T_{ref}}, \tag{2}$$

where $C_1$ and $C_2$ represent the model's shifting constants.

The master curves were modeled using the 2S2P1D model (2 springs, 2 parabolic elements, and 1 dashpot), as proposed by [61]. The model parameters are defined by Equation (3).

$$G^*(\omega) = G_\infty + \frac{G_0 - G_\infty}{1 + \delta(i\omega\tau)^{-k} + (i\omega\tau)^{-h} + (i\omega\beta\tau)^{-1}}, \tag{3}$$

where $G_\infty$ is the static modulus as $\omega \rightarrow 0$ (MPa), $G_0$ is the glassy modulus as $\omega \rightarrow \infty$ (MPa), $k$ and $h$ are dimensionless constants of the parabolic elements, with $0 < k < h < 1$, $\beta$ is a dimensionless constant related to the linear dashpot viscosity, $\delta$ is a dimensionless shape constant, $\tau$ is the characteristic time, which varies only with temperature (s), and $\omega$ is the pulsation $2\pi f$ ($s^{-1}$).

2.2.3. Multi-Stress Creep and Recovery Test (MSCR)

Following the Superpave classification [58], the MSCR test was conducted at the PGH temperature and at temperatures corresponding to $-1$PGH and $+1$PGH, ensuring that the results were obtained at the same temperature for the evaluated samples. The samples were tested for their virgin/original and RTFO conditions.

The creep and recovery scheme comprised 20 cycles at 0.1 kPa and 10 cycles at 3.2 kPa. Each cycle lasts 10 s, with 1 s of shear deformation and 9 s of rest. The lower stress level (first 20 cycles) falls within the linear viscoelastic (LVE) domain, while the higher stress level (last 10 cycles) is in the damage domain. The lower stress level corresponds to light traffic, and the higher stress level corresponds to heavy traffic conditions.

The initial 10 cycles at 0.1 kPa serve for sample conditioning, while the last 10 cycles at 0.1 kPa and the following 10 cycles at 3.2 kPa are employed to derive key critical parameters from the test. These parameters include the asphalt binder's delayed elastic response, quantified by the percentage of recovery (%R), an evaluation of the material's resistance to permanent deformation through non-recoverable compliance ($J_{nr}$), and an assessment of the material's sensitivity to changes in stress level, determined by the percent difference in non-recoverable creep compliance between 0.1 kPa and 3.2 kPa ($J_{nrdiff}$).

### 3. Results and Discussions

*3.1. Physical Characterization*

3.1.1. Penetration

Figure 3 presents the average results from the penetration tests conducted on both virgin/original samples at 25 °C. It can be observed that the Warm Mix Asphalt (WMA) additive led to a reduction of approximately 12% in penetration when compared to WMA_Ref ($49\ 10^{-1}$ mm) as opposed to HMA_Ref ($56\ 10^{-1}$ mm). With more Reclaimed Asphalt Pavement (RAP) content, penetration decreased, indicating an increase in sample stiffness. Notably, RAP_Original exhibited significantly lower penetration, measuring only $7\ 10^{-1}$ mm, attributed to the aging resulting from its field history of field application. This aging effect influenced the reduction in penetration observed in the WMA-RAP samples (from 20% to 47%), with the reduction being directly proportional to the percentage of RAP added to the sample.

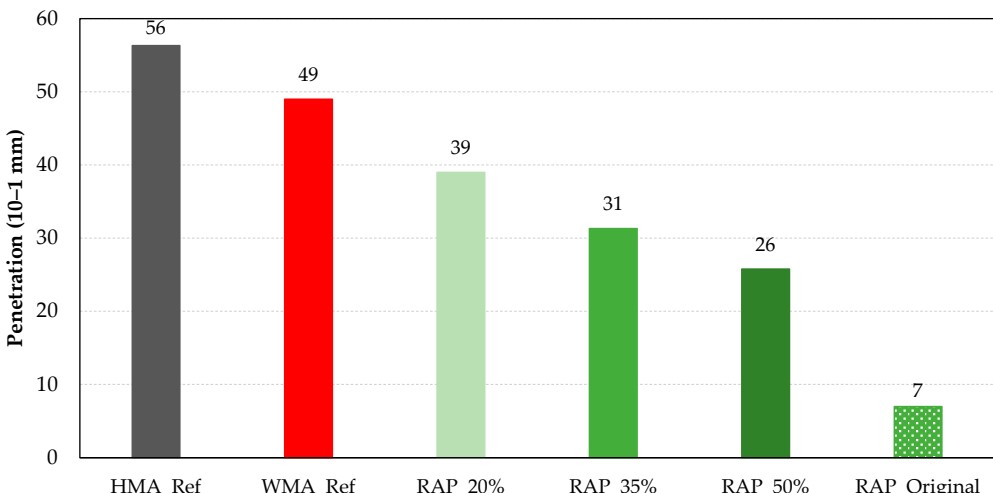

**Figure 3.** Penetration test results of virgin/original samples.

Regarding the retained penetration results after RTFO (Figure 4), it was evident that the WMA-RAP samples exhibited a higher percentage of retained penetration, less than 10%. On the other hand, the reference hot and warm samples experienced more significant penetration losses. These findings may indicate that the WMA samples, subjected to short-term aging at 133 °C, experienced lesser binder oxidation than the HMA sample. The RAP_Original sample, due to its advanced aging, presented a retained penetration of 86%. Thus, the WMA additive contributed to sustaining the physical properties after mixing and field compaction, maintaining similar retained penetration across the three studied WMA-RAP samples, regardless of the incorporated RAP ratio.

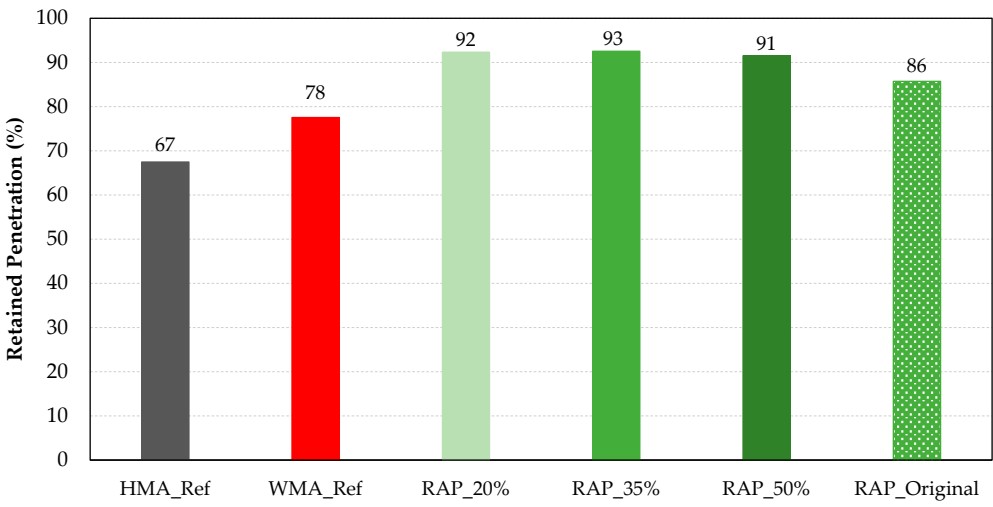

**Figure 4.** Retained penetration results.

The RAP contributed to maintaining penetration values based on the results obtained for WMA-RAP samples after RTFO. The results of the WMA-RAP samples indicated that this finding could be even more significant when the WMA was incorporated, as the additive contributed to the maintenance of the penetration of similar value and mitigated the aging effect. Also, these samples were less susceptible to aging than those without RAP and WMA additives.

From the penetration results of both WMA_Ref and RAP_Original samples, it was possible to estimate the penetration value of the WMA-RAP samples based on the quantity of RAP incorporated. However, further testing on various types of RAPs is necessary to substantiate this indication.

### 3.1.2. Softening Point

Figure 5 shows the softening point test results. The HMA_Ref sample exhibited a softening point of 49 °C, while the WMA_Ref sample had a slightly higher value at 51 °C. As the proportion of RAP in the WMA-RAP samples increased, the softening point also increased, ranging from 7% to 17%. This rise indicated enhanced rigidity in the samples. However, it's worth noting that this increase showed a slight decrease as the RAP content in the sample became more substantial.

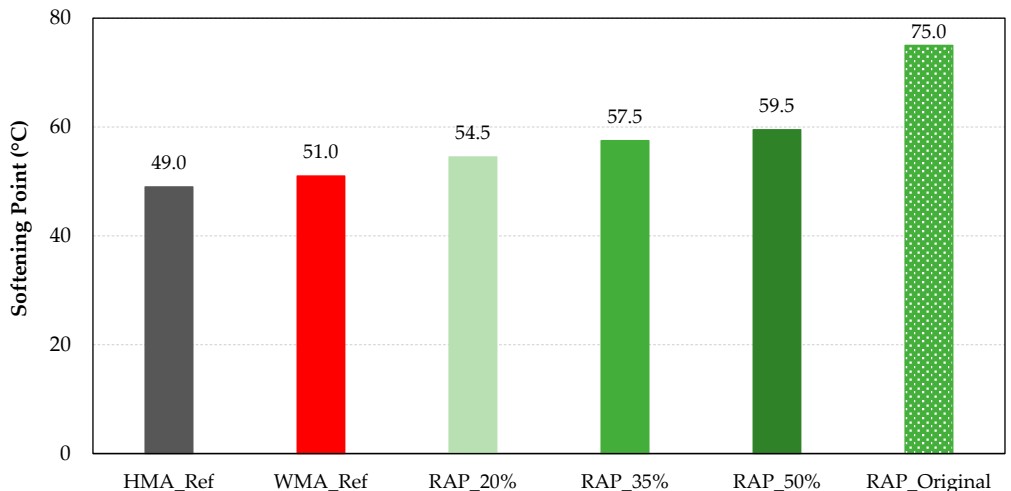

**Figure 5.** Softening point test data in virgin/original condition.

The Increment of Softening Point ($I_{SP}$), calculated as the difference between the softening point after short-term aging and the softening point of the samples in the virgin/original condition, is presented in Figure 6. HMA_Ref exhibited an increase of 4 °C, while RAP_Original presented an increase of 4.5 °C. WMA-RAP samples displayed a similar variation in the softening point after short-term aging, ranging from 2 °C to 2.5 °C, irrespective of the content of incorporated RAP. It could have occurred because the RFTO of WMA samples was set at 133 °C, indicating a reduced aging sensitivity of the WMA-RAP.

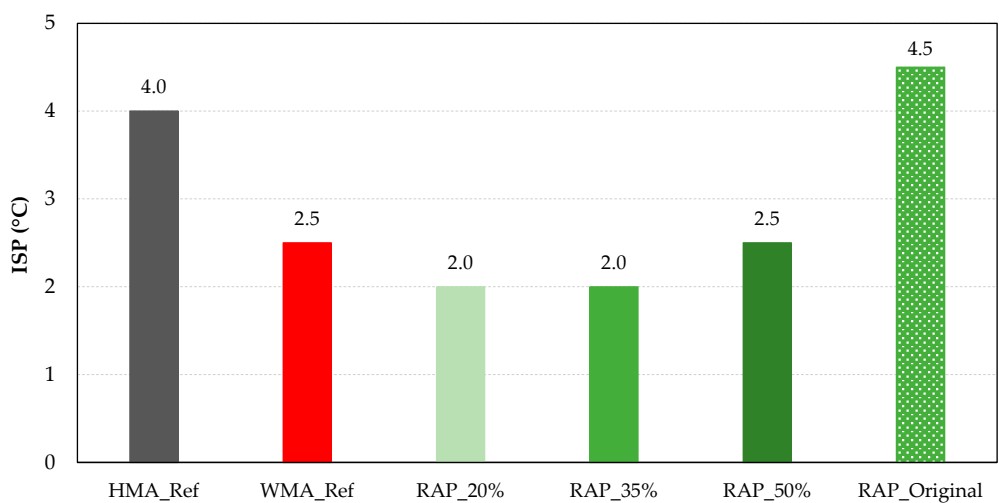

**Figure 6.** Increment of Softening Point test results.

Similar to the penetration test results, it was possible to indicate that the WMA-RAP samples were less susceptible to aging and that short-term aging at a temperature of 133 °C significantly improved the physical properties of the samples. Furthermore, the softening point results suggest that the addition of RAP and WMA additives can enhance the asphalt binder's resistance to permanent deformation. However, attention must be paid

to the increase in the softening point of the virgin/original samples due to the addition of RAP compared to the other tests to assess its influence on the performance of these asphalt binders.

### 3.2. Rheological Characterization

### 3.2.1. Viscosity

The apparent viscosity was measured to evaluate the material's workability at elevated temperatures. The tests were conducted using a Brookfield Viscometer equipped with spindle 21 at temperatures of 135 °C, 150 °C, and 177 °C, with rotations set at 20, 50, and 100 rpm, respectively. Figure 7 presents the obtained results.

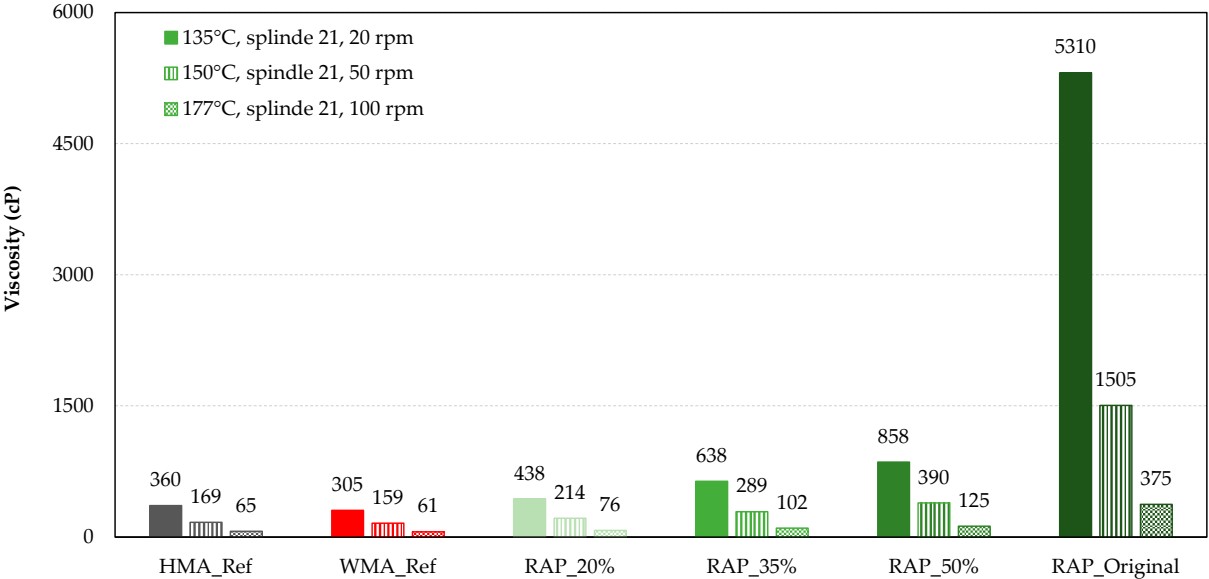

**Figure 7.** Apparent viscosity results.

The results reveal that the viscosity of HMA_Ref (360 cP) and WMA_Ref (305 cP) at 135 °C was quite similar, confirming the manufacturer's indication that the use of the WMA additive does not significantly influence viscosity. As expected, at 135 °C, the viscosity increased, ranging from 44% to 181% for WMA-RAP samples with increasing RAP content. However, it's important to note that the viscosity of RAP_Original (5310 cP) was notably higher than the other samples due to significant aging in the RAP. This sample would not meet the necessary viscosity limit for asphalt workability, which is limited to 3000 cP at 135 °C [57,58].

On the other hand, the other samples exhibit values below the limit, demonstrating that the use of the WMA additive with conventional virgin binder contributed to higher RAP contents while maintaining satisfactory workability, even with up to 50% RAP incorporation. Notably, the difference in viscosity among binders tended to decrease as the temperature increased.

Other studies have also concluded that WMA-RAP asphalt mixtures exhibit satisfactory workability [62]. Furthermore, according to Mejías-Santiago et al. [63], WMA-RAP mixtures showed adequate workability in comparison to HMA-RAP, with this characteristic decreasing as more RAP content was added to WMA mixtures.

The results of apparent viscosity after short aging are presented in Figure 8, and it was noted that all samples exhibit more elevated viscosity values after RTFO. Similarly to the virgin/original samples, viscosity increased with the addition of more significant RAP proportions. Table 5 presents the viscosity increments ($R_V$), revealing that samples without WMA additives exhibited a higher susceptibility to aging.

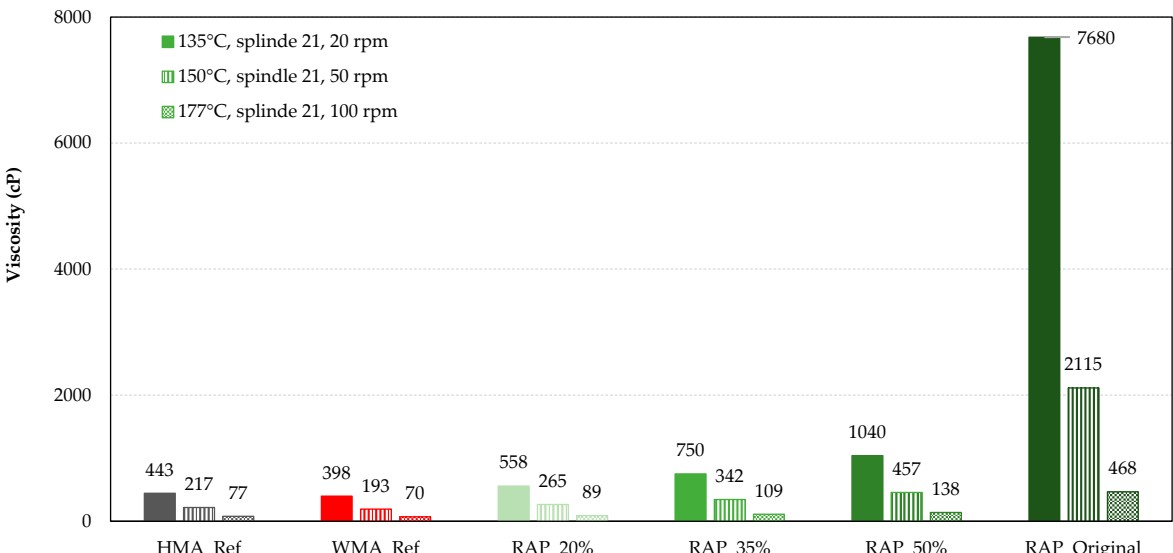

**Figure 8.** Apparent viscosity after RTFO.

**Table 5.** Increment of Viscosity data.

| Sample | $R_{V135°C}$ | $R_{V150°C}$ | $R_{V177°C}$ |
|---|---|---|---|
| HMA_Ref | 1.23 | 1.28 | 1.18 |
| WMA_Ref | 1.30 | 1.21 | 1.16 |
| RAP_20% | 1.27 | 1.24 | 1.18 |
| RAP_35% | 1.18 | 1.18 | 1.07 |
| RAP_50% | 1.21 | 1.17 | 1.10 |
| RAP_Original | 1.45 | 1.41 | 1.25 |

In a broader context, the behavior of WMA-RAP samples was consistent, with slight differences in stiffness enhancement compared to the virgin/original condition, with an increase ranging from approximately 7% to 27% across all tested temperatures. In contrast, for HMA_Ref and RAP_Original, this variation ranged from 18% to 45%. It is worth highlighting that $R_V$ decreased as the test temperature increased.

3.2.2. High-Temperature Performance Grade

The PGH and continuous PGH of the evaluated samples were determined. To comprehend the sample behavior due to RAP incorporation, the criteria were made to present the PGH results in the virgin/original and RTFO condition states, considering the Superpave requirements (Table 4).

To assess the conformity of the samples with the specification of performance-graded asphalt binder [57], the mass change after RTFO was determined. As previously mentioned, for samples with WMA additive, the test was conducted at 133 °C. The obtained results can be found in Table 6. Under the limitations of ASTM D2872 [52], the mass change of the samples must not exceed 1%. All samples exhibited mass losses below the specified limit in the standard. The most significant variation was observed in the RAP_Original sample, which displays a change of 0.11%. The remaining samples vary from 0.01% to 0.04%, indicating minimal mass alteration when subjected to the short-term aging process.

In the virgin/original conditions, as can be seen in Table 7, PGH remained unchanged for the HMA_Ref and WMA_Ref (PGH 64 °C) but increased with the increment of RAP in the WMA-RAP samples. Continuous PGH enhanced from 69.9 °C to 79.4 °C with RAP increasing from 20% to 50%. The RAP binder exhibited a PGH exceeding 94 °C. However, testing at even higher temperatures was not feasible due to the technical limitations of the DSR equipment. The results of PGH virgin/original samples were higher than those after RTFO aging.

**Table 6.** Mass change after RTFO.

| Asphalt Binder Sample | Temperature Test (°C) | Mass Change |
|:---:|:---:|:---:|
| HMA_Ref | 163 | 0.03% |
| WMA_Ref | 133 | 0.01% |
| RAP_20% | 133 | 0.03% |
| RAP_35% | 133 | 0.04% |
| RAP_50% | 133 | 0.03% |
| RAP_Original | 163 | 0.11% |

**Table 7.** PGH and continuous PGH data for samples in virgin/original and RTFO states.

| Sample | Virgin/Original | | RTFO | |
|:---:|:---:|:---:|:---:|:---:|
| | PGH (°C) | Continuous PGH (°C) | PGH (°C) | Continuous PGH (°C) |
| HMA_Ref | 64 | 64.0 | 58 | 62.3 |
| WMA_Ref | 64 | 64.2 | 58 | 60.7 |
| RAP_20% | 64 | 69.9 | 64 | 65.8 |
| RAP_35% | 70 | 74.6 | 70 | 69.9 |
| RAP_50% | 76 | 79.4 | 70 | 75.2 |
| RAP_Original | 94 * | 96.8 * | 94 * | 95.9 * |

* Estimated data, as the working limit of the DSR used in the research was 94 °C.

An improvement can be highlighted by analyzing the PGH results of the WMA-RAP samples after RTFO (Table 7). The samples with a 20% RAP increment showed a rise of 1PGH (64 °C), while those with 35% and 50% RAP content exhibited an increase of 2PGHs (70 °C) compared to the WMA_Ref (58 °C). The RAP_50% approached a PGH 76 grade. In terms of continuous PGH, there was an increase of 4.1 to 5.3 °C for WMA-RAP samples with each incremental percentage of RAP.

Comparing continuous PGH results for binders after RTFO with virgin/original samples, it was observed that samples subjected to short-term aging experienced a more substantial reduction in performance when the WMA additive was introduced. HMA_Ref and RAP_Original presented the slightest variation in continuous PGH, 1.7 °C and 0.9 °C, respectively. Conversely, the WMA_Ref presented a 3.5 °C difference, and the WMA-RAP samples obtained a difference of more than 4 °C (4.1 °C for RAP_20%, 4.7 °C for RAP_35%, and 4.2 °C for RAP_50%). This might imply that the WMA additive may have influenced this parameter, as including RAP does not necessarily represent an increase or decrease in this difference. Due to the inability to test PG at a temperature of 100 °C, the continuous PGH of the RAP_Original sample was an estimation.

Limited data are available regarding the high-temperature characteristics of WMA-RAP samples. However, some studies have shown promising results. For instance, Dong et al. [64] found that WMA-RAP samples with foam asphalt exhibited superior performance at high temperatures and greater temperature sensitivity compared to WMA alone. Similarly, Yu et al. [65] demonstrated that adding RAP increases the failure temperature of WMA samples with foam asphalt. Additionally, Almeida Junior et al. [66] observed an increase in PGH with greater binder aging. These findings suggest that WMA-RAP samples may have advantages in high-temperature performance.

In summary, the results of PGH suggest that the addition of WMA with 20% RAP has a minimal impact on the binder's performance, with no deterioration compared to the reference sample. This implies that it is feasible to produce WMA-RAP binders that offer economic and sustainable advantages for paving without compromising high-temperature pavement performance compared to conventional binders. Moreover, incorporating higher RAP contents, such as 35% and 50%, can potentially enhance performance even further.

### 3.2.3. Linear Viscoelastic Characterization—Master Curves

The rheological properties describe the material's behavior at different temperatures and loading frequencies. This study tested virgin/original samples and those subjected to RTFO. From TTSP, master curves for $|G^*|$ and δ were constructed for a $T_{Ref}$ of 20 °C. The translation factors ($a_T$) and the constants of the Williams-Landel-Ferry (WLF) model are presented in Table 8. The acquired data were fitted to the 2S2P1D rheological model, and the corresponding parameters are listed in Table 9.

**Table 8.** Master curve parameters.

| Samples | log (aT) | | | | | | WLF | |
|---|---|---|---|---|---|---|---|---|
| | 5 | 20 | 35 | 45 | 60 | 75 | C1 | C2 |
| Virgin/Original | | | | | | | | |
| HMA_Ref | 2.40 | 0.00 | −1.90 | −3.05 | −4.05 | −4.90 | 14.99 | 113.20 |
| WMA_Ref | 2.65 | 0.00 | −1.95 | −2.85 | −3.90 | −4.75 | 14.66 | 114.70 |
| RAP_20% | 2.25 | 0.00 | −2.05 | −3.25 | −4.45 | −5.35 | 15.96 | 109.09 |
| RAP_35% | 2.30 | 0.00 | −2.00 | −3.30 | −4.45 | −5.40 | 16.15 | 109.47 |
| RAP_50% | 2.25 | 0.00 | −1.95 | −3.25 | −4.43 | −5.45 | 16.56 | 112.10 |
| RAP_Original | 2.45 | 0.00 | −1.85 | −3.15 | −4.50 | −5.65 | 16.74 | 107.92 |
| RTFO | | | | | | | | |
| HMA_Ref | 2.50 | 0.00 | −2.00 | −3.10 | −4.15 | −5.00 | 15.68 | 117.46 |
| WMA_Ref | 2.45 | 0.00 | −1.90 | −2.90 | −3.90 | −4.70 | 15.09 | 121.60 |
| RAP_20% | 2.25 | 0.00 | −2.00 | −3.15 | −4.20 | −5.15 | 15.97 | 115.51 |
| RAP_35% | 2.25 | 0.00 | −2.00 | −3.20 | −4.35 | −5.30 | 15.83 | 109.26 |
| RAP_50% | 2.25 | 0.00 | −1.95 | −3.25 | −4.45 | −5.45 | 16.56 | 112.10 |
| RAP_Original | 2.35 | 0.00 | −1.95 | −3.15 | −4.60 | −5.80 | 16.93 | 105.57 |

**Table 9.** 2S2P1D model parameters.

| Samples | Parameters | | | | | | |
|---|---|---|---|---|---|---|---|
| | $G_{00}$ (Pa) | $G_0$ (Pa) | k | h | δ | $\tau_0$ | β |
| Virgin/Original | | | | | | | |
| HMA_Ref | 0 | $1.70 \times 10^9$ | 0.25 | 0.65 | 9.0 | $2.50 \times 10^{-5}$ | 60 |
| WMA_Ref | 0 | $1.70 \times 10^9$ | 0.30 | 0.68 | 8.0 | $2.50 \times 10^{-5}$ | 40 |
| RAP_20% | 0 | $4.00 \times 10^9$ | 0.28 | 0.65 | 13.0 | $2.50 \times 10^{-5}$ | 120 |
| RAP_35% | 0 | $4.50 \times 10^9$ | 0.23 | 0.65 | 15.0 | $3.50 \times 10^{-5}$ | 180 |
| RAP_50% | 0 | $5.00 \times 10^9$ | 0.23 | 0.66 | 17.0 | $5.50 \times 10^{-5}$ | 200 |
| RAP_Original | 0 | $9.00 \times 10^9$ | 0.20 | 0.60 | 25.0 | $1.00 \times 10^{-4}$ | 1400 |
| RTFO | | | | | | | |
| HMA_Ref | 0 | $3.80 \times 10^9$ | 0.24 | 0.65 | 14.0 | $1.30 \times 10^{-5}$ | 110 |
| WMA_Ref | 0 | $2.20 \times 10^9$ | 0.27 | 0.65 | 9.0 | $2.00 \times 10^{-5}$ | 50 |
| RAP_20% | 0 | $4.00 \times 10^9$ | 0.24 | 0.64 | 14.0 | $2.00 \times 10^{-5}$ | 130 |
| RAP_35% | 0 | $4.50 \times 10^9$ | 0.22 | 0.65 | 16.0 | $3.00 \times 10^{-5}$ | 210 |
| RAP_50% | 0 | $5.00 \times 10^9$ | 0.22 | 0.65 | 18.0 | $5.00 \times 10^{-5}$ | 260 |
| RAP_Original | 0 | $9.00 \times 10^9$ | 0.19 | 0.58 | 26.0 | $1.20 \times 10^{-4}$ | 3300 |

The master curves of $|G^*|$ and δ for the virgin/original binders are presented in Figures 9 and 10, respectively. These curves reveal that the dynamic modulus increased as the frequency rose. This increase can be attributed to reduced exposure to loading at higher frequencies and the limited time for reversible deformations to occur. At higher frequencies, the material exhibits predominantly elastic deformations, resulting in peak modulus values. In contrast, at lower frequencies, viscoelastic deformations become more

pronounced due to the longer loading times, leading to lower dynamic modulus values and higher phase angle values.

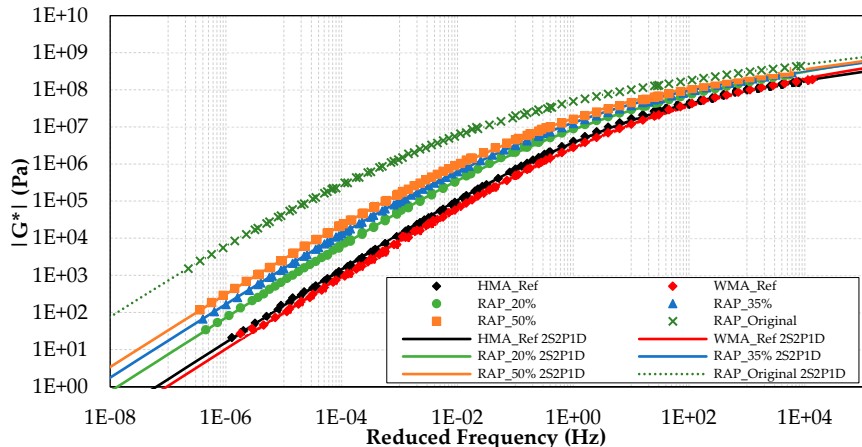

**Figure 9.** Master curves of |G*| under virgin/original conditions.

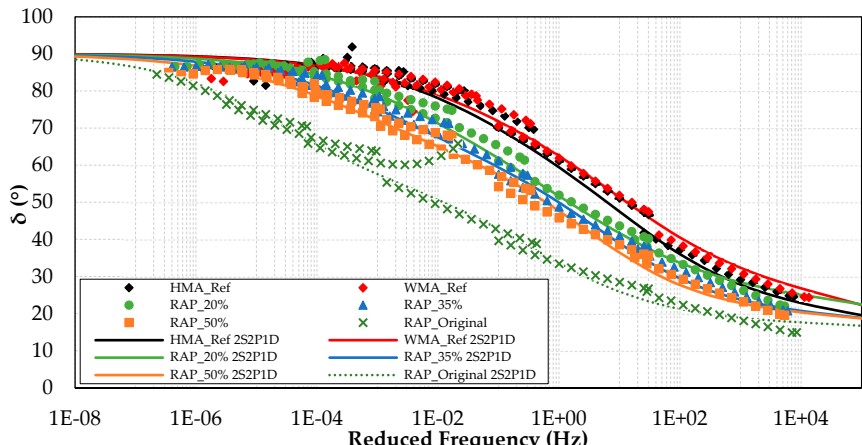

**Figure 10.** Master curves of δ at virgin/original condition.

Moreover, Figure 9 illustrates that the WMA_Ref sample obtained lower stiffness (|G*|) compared to the HMA_Ref sample across the entire frequency spectrum. This difference is attributed to the influence of the WMA additive on rheological properties, making the material less stiff. On the other hand, the master curves for the WMA-RAP samples showed increased stiffness as the RAP content increased. This stiffening effect was more pronounced with higher RAP content and increased loading frequency, ultimately enhancing the material's stability at elevated temperatures. When analyzing stiffness at $T_{ref}$ 20 °C and 10 Hz, it is possible to measure that the stiffness increased by 247%, 314%, and 380%, respectively, for samples containing 20%, 35%, and 50% of RAP. The RAP_Original exhibited substantially higher stiffness than the other samples and a lower phase angle. This can be attributed to oxidation and aging that occurred in the field.

Figure 10 reveals that the lowest values of δ were observed in the RAP_Original sample, followed by samples with the highest to lowest increments of RAP and the reference binders. Under the conditions of $T_{ref}$ 20 °C and 10 Hz, the phase angle decreased by 16%, 21%, and 26% for samples containing 20%, 35%, and 50% of RAP, respectively. At lower frequencies, less than $1 \times 10^{-4}$, the samples presented a similar behavior, approaching a purely viscous response (δ = 90°), except for the RAP reference sample, which exhibited a lower value. However, at intermediate and high frequencies, the addition of RAP led to a more elastic behavior rather than a tendency toward viscous flow. This resulted in increased energy

storage within the material, allowing these materials to accumulate stress more rapidly during repeated deformation processes.

The master curves for the aged samples are shown in Figures 11 and 12. When comparing the results of the virgin/original and aged samples, it can be noted that the curves exhibit greater stiffness, leading to an overall increase in the |G*| value for all binders (Figure 11) across the entire frequency spectrum. Relative to the δ value (Figure 12), there is a decrease in the phase angle due to the short-term aging process, indicating a higher degree of aging in the samples, which can be attributed to the RTFO procedure.

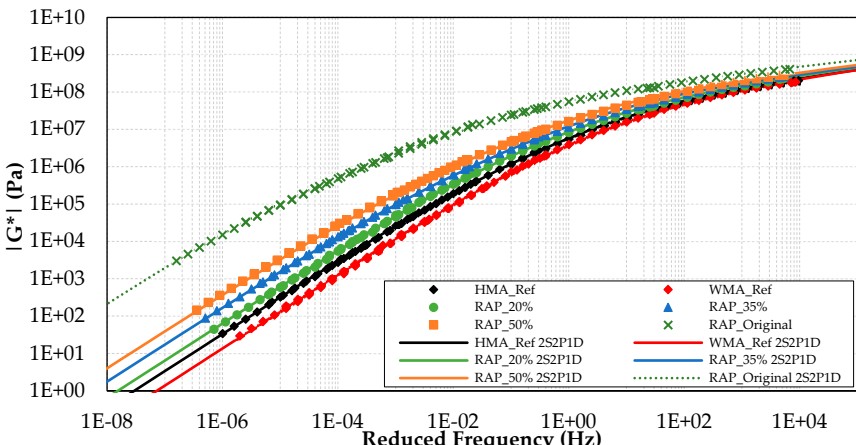

**Figure 11.** Master curves of |G*| after RTFO.

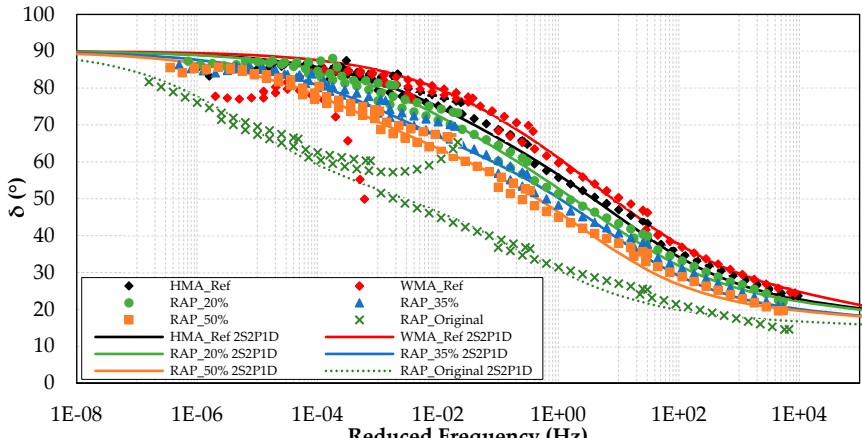

**Figure 12.** Master curves of δ after RTFO.

Furthermore, from the master curves, changes due to the aging effect in |G*| and δ were observed by the magnitude variations along the reduced frequency range under consideration. The rheological alterations occurred non-proportionally concerning the frequency spectrum and temperature. Regarding |G*|, the lower frequencies and higher temperatures undergo relatively more significant changes compared to the higher frequencies and lower temperatures for all evaluated samples. In contrast, for δ, a shift of nearly the entire curve towards lower phase angle values was stated, with more substantial alterations at higher frequencies and lower temperatures. These changes were more pronounced in the reference samples (HMA_Ref, WMA_Ref, and RAP_Original), indicating that the WMA-RAP samples were less sensitive to aging, likely due to the lower temperature of the RTFO tests.

### 3.3. Permanent Deformation

Multi-Stress Creep and Recovery (MSCR) Test

In order to predict permanent deformation, the MSCR test was conducted, and the parameters $J_{nr}$ and R% were calculated. The test was performed at different temperatures for each sample, ranging from 58 °C to 94 °C, as presented in Table 10, ensuring the same tested temperatures for comparison.

**Table 10.** MSCR test data after RTFO.

| Sample | PGH | PGH Classification MSCR | Parameter | Test Temperature (°C) | | | | | |
|---|---|---|---|---|---|---|---|---|---|
| | | | | 58 | 64 | 70 | 76 | 88 | 94 |
| HMA_Ref | 58 | 58S | $J_{nr0.1}$ (kPa$^{-1}$) | 2.663 | 6.005 | 12.355 | - | - | - |
| | | | $R_{0.1}$ (%) | 0.000 | 0.000 | 0.000 | - | - | - |
| | | | $J_{nr3.2}$ (kPa$^{-1}$) | 2.813 | 6.254 | 12.863 | - | - | - |
| | | | $R_{3.2}$ (%) | 0.000 | 0.000 | 0.000 | - | - | - |
| | | | $J_{nrdiff}$ (%) | 5.606 | 4.160 | 4.115 | - | - | - |
| WMA_Ref | 58 | 58S | $J_{nr0.1}$ (kPa$^{-1}$) | 3.506 | 7.789 | 16.137 | - | - | - |
| | | | $R_{0.1}$ (%) | 0.000 | 0.000 | 0.000 | - | - | - |
| | | | $J_{nr3.2}$ (kPa$^{-1}$) | 3.666 | 8.162 | 16.770 | - | - | - |
| | | | $R_{3.2}$ (%) | 0.000 | 0.000 | 0.000 | - | - | - |
| | | | $J_{nrdiff}$ (%) | 4.569 | 4.788 | 3.925 | - | - | - |
| RAP_20% | 64 | 64S | $J_{nr0.1}$ (kPa$^{-1}$) | - | 3.509 | 7.760 | 16.220 | - | - |
| | | | $R_{0.1}$ (%) | - | 0.000 | 0.000 | 0.000 | - | - |
| | | | $J_{nr3.2}$ (kPa$^{-1}$) | - | 3.695 | 8.175 | 16.906 | - | - |
| | | | $R_{3.2}$ (%) | - | 0.000 | 0.000 | 0.000 | - | - |
| | | | $J_{nrdiff}$ (%) | - | 5.309 | 5.350 | 4.231 | - | - |
| RAP_35% | 70 | 64H | $J_{nr0.1}$ (kPa$^{-1}$) | - | 1.929 | 4.412 | 9.432 | - | - |
| | | | $R_{0.1}$ (%) | - | 0.000 | 0.000 | 0.000 | - | - |
| | | | $J_{nr3.2}$ (kPa$^{-1}$) | - | 2.003 | 4.662 | 10.036 | - | - |
| | | | $R_{3.2}$ (%) | - | 0.000 | 0.000 | 0.000 | - | - |
| | | | $J_{nrdiff}$ (%) | - | 4.896 | 5.675 | 6.405 | - | - |
| RAP_50% | 70 | 64V ou 70H | $J_{nr0.1}$ (kPa$^{-1}$) | - | 0.843 | 2.085 | 4.651 | - | - |
| | | | $R_{0.1}$ (%) | - | 0.000 | 0.000 | 0.000 | - | - |
| | | | $J_{nr3.2}$ (kPa$^{-1}$) | - | 0.913 | 2.216 | 4.968 | - | - |
| | | | $R_{3.2}$ (%) | - | 0.000 | 0.000 | 0.000 | - | - |
| | | | $J_{nrdiff}$ (%) | - | 8.220 | 6.265 | 6.814 | - | - |
| RAP_Original | 94 * | 88H ou 94S | $J_{nr0.1}$ (kPa$^{-1}$) | - | - | - | - | 1.467 | 3.199 |
| | | | $R_{0.1}$ (%) | - | - | - | - | 0.000 | 0.000 |
| | | | $J_{nr3.2}$ (kPa$^{-1}$) | - | - | - | - | 1.552 | 3.257 |
| | | | $R_{3.2}$ (%) | - | - | - | - | 0.000 | 0.000 |
| | | | $J_{nrdiff}$ (%) | - | - | - | - | 5.804 | 1.802 |

* Estimated data, as the working limit of the DSR used in the research was 94 °C.

Table 10 results showed that in all samples, the R% value was zero, confirming that the RAP binder originates from a conventional binder, similar to the virgin binder used in this study. Therefore, since R% represents a measure of recovery capacity typically found in polymers, no recovery indicated that the studied asphalts do not contain any polymers in their composition and do not regain their properties when subjected to stresses and deformations in the MSCR test. Thus, the WMA-RAP samples exhibited compromised elastic responses like the reference samples, which already displayed a null %R value.

The samples exhibited high $J_{nr0.1}$ and $J_{nr3.2}$ values (Table 10), attributed to the potentially poor performance of the virgin binder. This was more evident when comparing the data from the HMA_Ref and WMA_Ref samples, which showed similar results for the tests conducted at temperatures ranging from 58 °C to 70 °C. However, it is worth highlighting that the $J_{nr3.2}$ values were higher for the warm sample, indicating its lower resistance to permanent deformation than the hot sample.

The WMA-RAP samples exhibited improved characteristics (Table 10) regarding permanent deformation, with the influence of RAP contributing to this behavior. As the RAP content increased, the $J_{nr3.2}$ values decreased, indicating enhanced resistance to higher traffic levels (Figure 13). Similar findings were reported by Zaremotekhases et al. [67], who demonstrated that samples with RAP, at both testing levels, exhibited lower $J_{nr}$ values than

those without RAP, indicating the stiffening effect of RAP binder. According to Yu et al. [65], increasing amounts of RAP binder lead to a decrease in the $J_{nr}$ value of WMA-RAP samples with foam asphalt and an increase in the %R value. Thus, a higher amount of RAP binder enhanced the asphalt's resistance to permanent deformation. The $J_{nrdiff}$ (%) values (Table 10) were within the acceptable range according to the standard, ranging from 1.8% to 8.2%, which is lower than the 75% limit [58], indicating minimal interference from the test levels of MSCR.

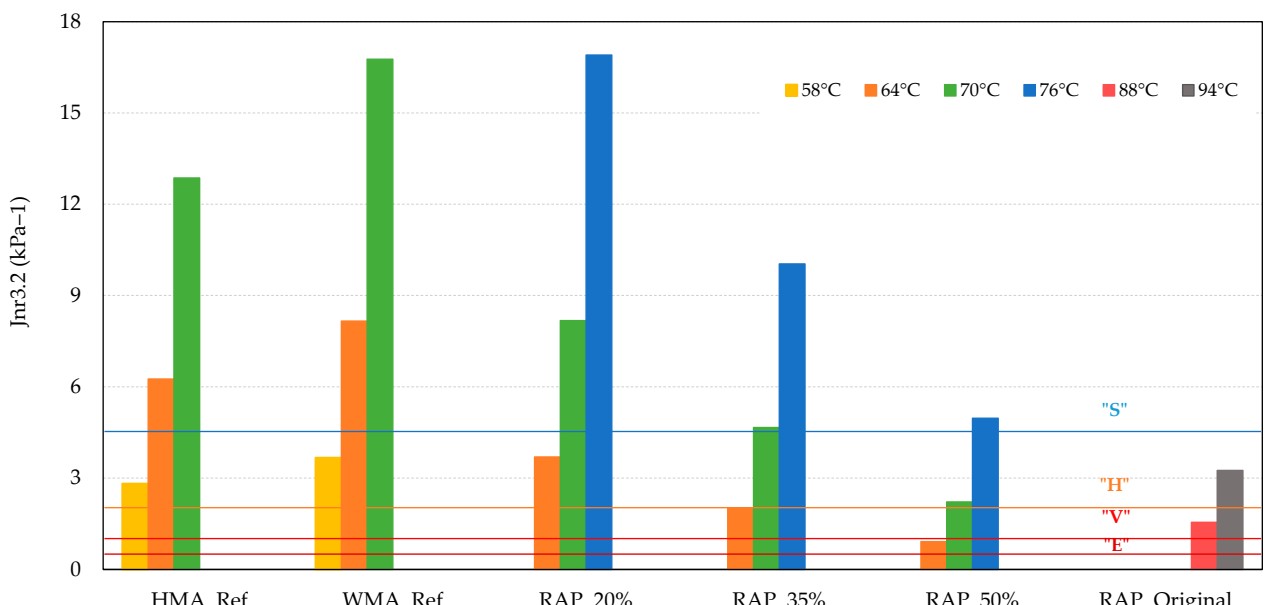

**Figure 13.** $Jnr_{3.2}$ data after RTFO with AASHTO M332 [58] limits.

According to AASHTO M332 [58], based on the value obtained for $J_{nr3.2}$, the material can be classified for Standard "S" traffic ($2.0 < J_{nr3.2} < 4.5$), Heavy "H" traffic ($1.0 < J_{nr3.2} < 2.0$), Very Heavy "V" traffic ($0.5 < J_{nr3.2} < 1.0$), and Extremely Heavy "E" traffic ($0.0 < J_{nr3.2} < 0.5$). As seen in Table 10, adding 20% RAP increased the PGH from 58S (reference samples) to 64S. The addition of 35% RAP, a material with the same PG as the RAP_20% sample, was achieved but classified for Heavy traffic (64H). When the RAP proportion was increased to 50%, the material was suitable for areas with PGH 64 for Very Heavy traffic (64V). Additionally, the RAP_50% sample met PGH 70H, indicating its ability to withstand PGH 70 conditions for Heavy traffic. The RAP_Original binder exhibited high performance with low $J_{nr3.2}$ values, which could be suitable for situations with PGH 88H or even 94S. Considering that the adopted virgin binder sample is of the conventional type, this significant improvement in permanent deformation with WMA-RAP samples could cover a wide range of PGH levels and traffic conditions on Brazilian highways.

Due to the elevated $J_{nr3.2}$ values obtained in the results, the materials did not perform well in terms of permanent deformation at certain temperatures, mainly the virgin binder. On the other hand, testing the asphalt mixture is essential since the aggregate matrix plays a crucial role in permanent deformation. Thus, it is essential to evaluate the behavior of these materials through mixture production and mechanical tests to understand their performance.

Corroborating with the results obtained for softening point and PGH, it is evident that there is an improvement in the performance concerning permanent deformation with the addition of WMA and RAP. The greater the increment of recycled pavement material, the more pronounced the improvement in the behavior of the asphalt binder, which aligns with the resource-efficient and environmentally conscious principles of "Net Zero" policies. This study demonstrated that WMA-RAP blends present a viable alternative in locations requiring enhanced pavement performance under high-temperature conditions.

As a result, these findings highlight the crucial role of WMA-RAP blends in mitigating carbon emissions and minimizing the broader environmental impact, perfectly aligning with the core tenets and objectives of "Net Zero" policies focused on sustainability.

## 4. Conclusions

This study aimed to investigate Warm Mix Asphalt (WMA) technology with Reclaimed Asphalt Pavement (RAP) addition and assess how these materials interact at the asphalt binder level. The economic benefits and sustainability aspects, considering greenhouse gas reduction policies and the rising global average temperature [20–22], make these types of materials promising solutions to be widely adopted in the coming years [29–31].

Moreover, it aligns with the United Nations' Sustainable Development Goal 9, which centers on industry, innovation, and infrastructure. Specifically, this research contributed to the development of resilient infrastructure by creating advanced materials and technologies for longer-lasting and sustainable critical structures. Furthermore, it promoted sustainable practices within the construction industry and presented innovative solutions to infrastructure challenges, fostering innovation in the field. These points underscore the significance of the study.

Compared to reference samples, the research evaluated WMA-RAP samples and their physical, rheological, and permanent deformation behavior characteristics. The results demonstrated that increasing the RAP content significantly enhanced sample performance, highlighting the potential of WMA-RAP as a sustainable paving alternative.

The main findings of this study can be summarized as follows:

- The interaction between the virgin binder, WMA additive, and RAP binder showed no negative behavior in high-temperature tests.
- Physical assessment indicated a decrease in penetration (20% to 47%) and an increase in softening point (7% to 17%), including 20% to 50% RAP in the samples.
- Apparent viscosity rose with higher RAP content at 135 °C (from 44% to 181%). However, the workability of all WMA-RAP samples remained suitable, staying below 3000 cP at 135 °C.
- Samples with added RAP displayed increased PGH values compared to reference samples: 1 PGH at 20% RAP and 2 PGHs at 35% and 50% RAP. Additionally, continuous PGH analysis showed a temperature rise of 4.1 °C to 5.3 °C with each RAP content increase in WMA-RAP samples.
- Master curves showed increased material stiffness and decreased phase angle, indicating aging when RAP binder was added. However, WMA-RAP samples displayed reduced sensitivity to aging across the entire frequency spectrum compared to reference samples, likely due to the RTFO being conducted at 133 °C.
- MSCR results showed that increasing the RAP binder content improved resistance to permanent deformation in WMA-RAP samples. The $J_{nr3.2}$ value decreased as higher RAP contents were added, indicating the ability to withstand heavier traffic loads for the 64S, 64H, and 64V/70H categories, respectively, in samples with 20%, 35%, and 50% RAP.
- WMA samples with up to 50% RAP exhibited satisfactory performance regarding permanent deformation. Despite increased aging due to higher RAP content, they showed higher PGH values, improved physical and rheological properties, and enhanced performance. Therefore, WMA-RAP samples demonstrated satisfactory behavior, even with significant RAP.

The findings indicate that permanent deformation resistance is not a barrier to utilizing WMA-RAP. This allows for lower-temperature mix production, reducing energy consumption, enhancing sustainability, and utilizing recycled materials. Nevertheless, chemical analysis is required for a more comprehensive understanding of material interactions. Additionally, evaluating fatigue resistance, water susceptibility, and testing other RAP binders should be considered for future research validation.

**Author Contributions:** Conceptualization, K.A.B., L.P.T. and L.P.S.; methodology, K.A.B.; validation, L.P.T. and L.P.S.; formal analysis, K.A.B.; investigation, K.A.B.; resources, L.P.T.; writing—original draft preparation, K.A.B.; writing—review and editing, L.P.T. and L.P.S.; visualization, K.A.B.; supervision, L.P.T.; project administration, L.P.T.; funding acquisition, L.P.T. and L.P.S. All authors have read and agreed to the published version of the manuscript.

**Funding:** This research received external funding in the form of a scholarship awarded to the first author. The scholarship was generously provided by the National Council for Scientific and Technological Development (CNPq), under Grant No. 432588/2016-7, and by the Coordination of Higher Level Staff Improvement—Brazil (CAPES) with the Finance Code 001.

**Institutional Review Board Statement:** Not applicable.

**Informed Consent Statement:** Not applicable.

**Data Availability Statement:** All data are shown in the manuscript.

**Acknowledgments:** The authors would like to thank CNPq/BRASIL, CAPES/BRASIL, and CBB Asfaltos for donating Neat Asphalt and WMA additive.

**Conflicts of Interest:** The authors declare no conflict of interest.

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
