# Peer review of "Physical, Rheological, and Permanent Deformation Behaviors of WMA-RAP Asphalt Binders"

_sustainability, doi:10.3390/su151813737_

Round 1

Reviewer 1 Report

Thanks for submission to MDPI.

Abstract

WMA------Pls use full name in each section once than use abbreviations

RAP------Pls use full name in each section once than use abbreviations

Line 13 behavior characteristics - Can we delete behavior pls revisit?

Line 16-17 A chemical surfactant-type 16 WMA additive was employed--- How?

Line 17 and onwards indicated that (1) ----pls remove numbering and follow MDPI format

Line 18 -19 increased stiffness (how much ???) and a lower phase angle (how much ???), along with reduced penetration (how much ???) and an increased softening point (how much??) 

Line 19, 20, 21 RAP content rises (how much ???) , (2) WMA-RAP samples are less sensitive (how much ???) to short-term aging effects, (3) a significant improvement (how much ???) in PGH is observed with the addition of RAP, and (4) WMA samples containing up to 50% RAP can adequately withstand permanent deformation (how much ???).

Can you add a line at the end explaining the industrial/social application of your research?

Keywords: Kindly add few from the section and special issue

Sustainability | Special Issue : Research on Sustainable Infrastructure Construction and Green Building Construction Materials (mdpi.com)

Introduction, Materials and Methods, Results and discussion

References and citation pls use MDPI guidelines

Line 35  state-of-the-art knowledge like -----------------------

Line 41 Studies have demonstrated --- which studies like recent/previous/research ?

Line 56 Considering the "Net Zero" policies ? Reference pls

WMA------Pls use full name in each section once than use abbreviations

RAP------Pls use full name in each section once than use abbreviations

Line 59 One such technology is Warm Mix 59 Asphalt (WMA), which reduces the manufacturing and compaction temperatures of samples, thus conferring environmental benefits such as reduced energy consumption during 61 production and, consequently, diminished fuel usage and emissions (Reference pls)

Line 100 It was evident that authors predominantly underscored the same benefits of WMA- 100 RAP, highlighting its significant environmental and economic potential----- why kindly expand?

Line 131 Can you pls enlist some properties of RAP?

Line 135 The incorporated WMA additive was of the chemical surfactant type, Evotherm® P25 Can you pls introduce some properties of WMA. 

Figure 1 can you pls add tests performed under the level of the physical, rheological and permanent deformation boxes?

Figures 8 and 9 the text of the legends not clear?

Figure 10 Pls increase the font of the index lines labels

Can authors pls explain the content in material and methods and results under following sequence as given in title:

1. Physical

2. rheological

3. permanent deformation

Presently the contents are intermixed in different sections. 

How can be the research support the SDG of the UNO you can introduce some discussion?

Line 496 The economic benefits and sustainability aspects? Can you add an analysis about this comparison?

Conclusions:

The interaction of virgin binder with WMA additive and RAP binder does not show negative behavior in the presented tests like ------

From the physical characteristics evaluated, it can be concluded that was a reduction in penetration (from ----- to  -------) and an increase in softening point (from ----- to  -------) with the addition of RAP in the samples up to ---------- %.

Also, WMA-RAP samples were less sensitive (from ----- to  -------) to short- term aging effects.

The apparent viscosity increased (from ----- to  -------) with higher RAP content (????)

however, the workability of all WMA-RAP samples remained adequate (?????)

An increase (from ----- to  -------)  in PGH for samples with RAP addition compared to reference samples was observed. 

References:

Pls observe the MDPI style and add few more references from the MDPI

Few small corrections in grammar and formatting style are requested. Thanks

Author Response

Dear Reviewer#1,

I want to express my sincere gratitude for taking the time to review our article. Your insightful comments and suggestions have been invaluable in enhancing the quality of our work. We have incorporated all applicable suggestions and made the necessary corrections based on your feedback.

Attached herewith, you will find our detailed responses to the specific points raised in your review. We hope that these responses adequately address your concerns and meet your expectations.

Once again, I would like to extend my thanks for your time and dedication to this review process. Your contributions have played a pivotal role in refining our article, and we are hopeful that it now meets the criteria for acceptance.

Should you have any further comments or require any additional information, please do not hesitate to reach out. We greatly value your input and are committed to making any further improvements as necessary.

Kind regards,

Kátia Aline Bohn

Reviewer 2 Report

The paper is looking technically sound but still some major corrections are required before it's final publication. 

1. Use the full form first time then you can use short form.....Like......WMA-RAP in the abstract section.....Improve this in the whole manuscript. 

2. Present the material photograph in the material section. Also, the physical properties of material used. 

3. Show the sustainability aspect in the introduction part. For that used the follow article:

-https://doi.org/10.1016/j.jclepro.2022.132969
-https://doi.org/10.1016/j.istruc.2023.05.094
4. Explain that why interaction of virgin binder with WMA additive and RAP binder does  not show negative behavior with the help of suitable literature. 
5. The readability of master curve is quite difficult....So Improve the quality of graph. 

6. Authors suggested to reduce the conclusion size in terms of number of words. 

7. Present the single paragraph with crisp detail summary of the research article in the conclusion section. 

Spelling and Grammatical error have to be rectified throughout the whole manuscript. 

Author Response

Dear Reviewer#2,

I want to express my sincere gratitude for taking the time to review our article. Your insightful comments and suggestions have been invaluable in enhancing the quality of our work. We have incorporated all applicable suggestions and made the necessary corrections based on your feedback.

Attached herewith, you will find our detailed responses to the specific points raised in your review. We hope that these responses adequately address your concerns and meet your expectations.

Once again, I would like to extend my thanks for your time and dedication to this review process. Your contributions have played a pivotal role in refining our article, and we are hopeful that it now meets the criteria for acceptance.

Should you have any further comments or require any additional information, please do not hesitate to reach out. We greatly value your input and are committed to making any further improvements as necessary.

Kind regards,

Kátia Aline Bohn

Reviewer 3 Report

This study looked into the rutting performance of WMA mix containing RAP. Both technologies of reducing mixing temperature during production and/or paving and using RAP can provide environmental benefits. As no consensus has been reached from previous studies regarding the pavement performance of WMA-RAP, this study designed an experimental plan including traditional HMA, WMA no RAP, WMA with various contents of RAP. The short term aging was also considered as a study variable. Multiple laboratory tests at binder scale have been performed. This study will contribute to more in-depth knowledge about the effect of WMA, RAP and combined effect on rutting performance. The detailed comments are as follows:

(1). Line 56. A bit explanation is recommended on “Net Zero”.

(2) Some statements or summaries on cited references in the introduction part need to be verified:

     Line 63: “as the lower heating temperature of WMA enables higher RAP content”, can the authors provide references or explanation for lowering temperature enabling higher RAP content?

     Line 66 to 68: is 24 the correct reference for this statement? Reference 24 focused on the pollution emission comparison between half-warm mix and hot mix asphalt.

     Line 83 to 85: the statement seems to suggest that WMA technologies enhanced the workability of the new asphalt mixture with RAP. However, references (e.g. 31) only concluded that additives are the ones that helped to improve the workability for WMA with RAP.

     Line 101-104: After all the presented literature review, it seems “the resistance to permanent deformation of WMA-RAP mixtures remains a concern” (line 95). Why do the authors make the conclusion that “Reducing production and compaction temperatures of asphalt mixtures… yielding promising performance outcomes for new mixtures”?

(3) For Table 1, it was expected that the percentages of all sample compositions would add up to 100%. It is recommended that the WMA (should be WMA additive) can be removed from the table or be included in the same column with the Asphalt binder. This could ensure accurate and clear representation on the material portion.

(4) Have the authors considered PAV long term aging on binder material? In the introduction part (line 86-89), authors emphasized the importance of assessing the performance of WMA-RAP over the long term and the durability throughout service life is the focal point of this present research. Why was binder material only prepared under short term aging (RTFO)?

(5) It is recommended to reassess the conclusions made from Figure 3 (line 247 to 249). Comparing the retained penetration percent of WMA_Ref and the ones with RAP, it seems like RAP is the major contributor instead of WMA additive to remain the properties.

(6) Line 272-273: as HMA and WMA were RTFO aged at different temperatures, it would be difficult to make the conclusion of “reduction in the aging sensitivity of WAM compared to HMA and RAP_Original”. The same concern applies to statements in the Conclusions section. It would not be appropriate to state that WMA-RAP were less sensitive to short term aging, as WMA-RAP were subjected to less aging due to lower temperature.

(7) Line 445- 448: authors presented a study by Yu which showed opposite findings from this study, can the authors provide some explanation for this difference?

It is recommended to engage a native English speaker or professional proofreader for a thorough review of the content, focusing on grammatical errors and awkward phrasing. 

Author Response

Dear Reviewer#3,

I want to express my sincere gratitude for taking the time to review our article. Your insightful comments and suggestions have been invaluable in enhancing the quality of our work. We have incorporated all applicable suggestions and made the necessary corrections based on your feedback.

Attached herewith, you will find our detailed responses to the specific points raised in your review. We hope that these responses adequately address your concerns and meet your expectations.

Once again, I would like to extend my thanks for your time and dedication to this review process. Your contributions have played a pivotal role in refining our article, and we are hopeful that it now meets the criteria for acceptance.

Should you have any further comments or require any additional information, please do not hesitate to reach out. We greatly value your input and are committed to making any further improvements as necessary.

Kind regards,

Kátia Aline Bohn

Round 2

Reviewer 1 Report

Thanks for submitting the revised version.

Line 11 Net zero policy --- Kindly relate your research contribution towards it in somewhere in results / discussions sections. 

Line 16 The findings showed that the WMA-RAP combination resulted in increased stiffness (  from   to    ) and a reduced phase angle (from      to     ) ------ Kindly revisit to give some typical ranges

Line 21  64V or 70H --- what is this? Kindly express

Line 22 t lower temperatures------like ?

The reference citation in text should be as per MDPI format please

Line367 to 368 -----pls check the gaps?

Figures 9, 10, 11 and 12 the lot of same green shades can be changed with other colors for further clarity

Figure 13 pls use different bar colors

How findings contribute to sustainability pls add a paragraph before the conclusions section. 

How findings contribute to Net Zero pls add a paragraph before the conclusions section. 

How findings contribute towards cost can we introduce a cost analysis comparison in results and brief discussion? 

Regards

Better now few minor grammar correction are required. 

Author Response

Dear Reviewer#1,

Thank you for your valuable review. We've incorporated your suggestions and provided detailed responses to your comments in the attached document. Your input has been instrumental in improving our article. If you have more feedback or need additional information, please let us know.

Best regards,

Kátia Aline Bohn

Reviewer 2 Report

Authors have satisfactorily answered all the comments....So manuscript is suitable for the publication in the journal.

Author Response

Dear Reviewer#2,

We want to extend our heartfelt gratitude for your expertise and dedication during the review process. Your constructive feedback has significantly enhanced the quality of our work.

Best regards,

Kátia Aline Bohn

Reviewer 3 Report

The authors have made significant improvements to the manuscript and the reviewer's comments have been addressed comprehensively which is appreciated. 

The revised manuscript is much more coherent and easier to follow. Modified language helped clarify the ideas. Just a minor issue: it's common to spell out the acronym only the first time it shows up in text. You don't have to spell out RAP (an example) every time. 

Author Response

Dear Reviewer#3,

We want to extend our heartfelt gratitude for your expertise and dedication during the review process. Your constructive feedback has significantly enhanced the quality of our work.

Best regards,

Kátia Aline Bohn
